# An Energy-Efficient BJT-Based Temperature Sensor with ±0.8 °C (3*σ*) Inaccuracy from −50 to 150 °C

**DOI:** 10.3390/s22239381

**Published:** 2022-12-01

**Authors:** Chuyun Qin, Zhenyan Huang, Yuyan Liu, Jiping Li, Ling Lin, Nianxiong Tan, Xiaopeng Yu

**Affiliations:** 1Institute of VLSI Design, Zhejiang University, Hangzhou 310027, China; 2Beijing Smartchip Microelectronics Technology Company Limited, Beijing 100192, China; 3Vango Technologies Inc., Hangzhou 310053, China

**Keywords:** CMOS temperature sensor, energy-efficient, leakage, cascoded FIA

## Abstract

This article presents an energy-efficient BJT-based temperature sensor. The output of sensing front-ends is modulated by employing an incremental Δ-Σ ADC as a readout interface. The cascoded floating-inverter-based dynamic amplifier (FIA) is used as the integrator instead of the conventional operational transconductance amplifier (OTA) to achieve a low power consumption. To enhance the accuracy, chopping and dynamic element matching (DEM) are applied to eliminate the component mismatch error while β-compensation resistor and optimized bias current are used to minimize the effect of β variation. Fabricated in a standard 180-nm CMOS process, this sensor has an active area of 0.13 mm${}^{2}$. While dissipating only 45.7 μW in total, the sensor achieves an inaccuracy of ±0.8 °C (3σ) from −50 °C to 150 °C after one-point calibration.

## 1. Introduction

Unlike consumer electronics, the integrated circuits working in extreme conditions, such as automotive and industrial applications call for high-accuracy temperature sensors operating at much higher temperatures (>125 °C). As a result, off-chip thermistors and thermocouples are still mostly used because of the difficulty of fully-integrated CMOS implementation. Of course, many designs have been reported to investigate the possibility of on-chip temperature sensors in these kinds of extreme conditions. In CMOS technology, the temperature-sensing elements can be bipolar junction transistors (BJTs), MOSFETs, resistors, thermal diffusivity (TD), etc. [1]. The BJTs are characterized by long-term stability and high accuracy after one-point calibration [2,3,4]. Additionally, the substrate PNPs available in CMOS technology are not particularly sensitive to the mechanical stress caused by low-cost plastic packaging [5]; thus, BJTs are suitable for high-temperature applications. However, the leakage and saturation currents of CMOS components increase exponentially with temperature, which leads to significant temperature-sensing errors at high temperatures.

Taking other temperature-sensing elements into account at high temperatures, the increase in the cost caused by two-point calibration makes resistor-based temperature sensors less suitable for automotive applications [6]. CMOS temperature sensors based on the TD of silicon have the advantage of being insensitive to process deviations, as it is only related to the physical properties of the silicon material itself. However, the milliwatt-level power dissipation makes it less attractive [7,8].

There have already been some explorations about BJT-based temperature sensors with a wide temperature sensing range [9,10,11]. In order to achieve high accuracy at high temperatures, biasing current and readout circuit are optimized, and low leakage SOI processes are used in Ref. [9]. Some advanced technologies such as the 16 nm FinFET process is attempted in Ref. [10]. An on-chip low-cost heater-assisted voltage calibration (HA-VCAL) is used in Ref. [11] to further reduce calibration costs at the expense of system complexity. However, power consumption [9,10] and system complexity [11] are the drawbacks.

In this work, a BJT-based sensor that can achieve a good balance among power consumption, calibration cost and system complexity is presented. It achieves an inaccuracy of ±0.8 °C (3σ) from −50 to 150 °C after one-point calibration while consuming only 45.7 μW. The voltage signals provided by the sensing front-end are modulated by employing an incremental Δ-Σ ADC as a readout interface. The cascoded floating-inverter-based dynamic amplifier (FIA) is used as the integrator instead of the conventional operational transconductance amplifier (OTA). Since FIA has no need for static bias current, power consumption decreases, as well as system complexity. To enhance the accuracy, chopping and dynamic element matching (DEM) are applied to eliminate the component mismatch error while the β-compensation resistor and optimized bias current are used to minimize the effect of β variation. The temperature sensor is fabricated in a standard 180-nm CMOS process and has an active area of 0.13 mm${}^{2}$. This paper is organized as follows. In Section 2, the sensor’s operation principle is explained. The realization of the sensing front-end and readout circuits are depicted, respectively, in Section 3 and Section 4. An analysis of the measured results and a comparison with previous work are presented in Section 5. The conclusion is presented in Section 6.

## 2. Operation Principle

Temperature sensors based on BJTs utilize their characteristics. A CMOS smart temperature sensor includes two parts, which are analog front-end and readout interface. The analog front-end uses several bipolar transistors together with precision interface circuitry to extract voltage signals required for temperature measurement. The readout interface with an ADC provides a digital representation of their ratio [12].

Figure 1 illustrates how a PNP-based temperature sensor works [12]. In the core of the temperature sensor, two identical substrate PNPs ($Q_{1}$, and $Q_{2}$) are biased at different collector currents, which have a ratio of 1:*p*. The base–emitter voltage $V_{\mathrm{BE}}$ of $Q_{1}$ (or $Q_{2}$) is complementary to absolute temperature (CTAT) and can be expressed as follows:(1)$$V_{\mathrm{BE}}=\eta \frac{kT}{q}\ln\left(\frac{I_{C}}{I_{S}}\right)$$
where η≈ 1 is a non-ideality factor dependent on the process, *k* is the Boltzmann constant, *q* is the electron charge, *T* is the temperature in Kelvin, $I_{C}$ and $I_{S}$ are, respectively, the collector and saturation currents of the PNP. The $V_{\mathrm{BE}}$ corresponds to silicon’s bandgap voltage of about 1.2 V when extrapolated to 0 K (−273 °C). As a result of different collector currents, the difference between the two base–emitter voltages $\Delta V_{\mathrm{BE}}=V_{\mathrm{BE}2}-V_{\mathrm{BE}1}$ is proportional to absolute temperature (PTAT) and can be expressed as follows:(2)$$\Delta V_{\mathrm{BE}}=\eta \frac{kT}{q}\ln\left(p\right)$$
where *p* is the density ratio of their collector currents. When $V_{\mathrm{BE}}$ and $\Delta V_{\mathrm{BE}}$ are combined linearly, the resulting voltage is relatively constant. The resulting voltage $V_{\mathrm{REF}}=V_{\mathrm{BE}}+\alpha \cdot \Delta V_{\mathrm{BE}}$ also corresponds to the silicon bandgap voltage, where α is a constant.

By using an ADC, a digital temperature readout interface can be created. It measures temperature by digitizing $\alpha \cdot \Delta V_{\mathrm{BE}}$ with respect to $V_{\mathrm{REF}}$. The digital ratio μ is given by:(3)$$\mu =\frac{\alpha \cdot \Delta V_{\mathrm{BE}}}{V_{\mathrm{REF}}}=\frac{\alpha \cdot \Delta V_{\mathrm{BE}}}{V_{\mathrm{BE}}+\alpha \cdot \Delta V_{\mathrm{BE}}}$$
since $V_{\mathrm{REF}}$ is a voltage that is temperature-independent and relatively constant, μ will be PTAT. The final digital output $D_{\mathrm{out}\;}$ can be measured by linear scaling in degrees Celsius:(4)$$D_{\mathrm{out}\;}=A\cdot \mu +B$$
where *A* and *B* are constant coefficients.

There is a discovery that $V_{\mathrm{BE}}$ and $\Delta V_{\mathrm{BE}}$ include all the necessary temperature information. Consequently, using a relatively constant voltage $V_{\mathrm{REF}}$ as the reference voltage is not necessary [3]. As an alternative, the ADC can digitize $\Delta V_{\mathrm{BE}}$ with respect to $V_{\mathrm{BE}}$. The result can be $X=\gamma \cdot \Delta V_{\mathrm{BE}}/V_{\mathrm{BE}}$, where γ is a natural number. At the same time, the variation of η can be omitted as it is canceled out. The digital ratio μ can then be determined after formula deformation:(5)$$\mu =\frac{k\cdot X}{1+k\cdot X}$$
where *k* is a calibration parameter and can be calculated easily in a digital back-end; thus, it is a constant unaffected by process spread. It also can be seen as a mapping factor between *X* and digital ratio μ, which is variable when trimming temperature-sensing errors. This is much more convenient to adjust than analog techniques [3]. This method transfers signal processing from the analog to digital domain; therefore, power consumption, system complexity and area of the analog circuitry are reduced.

The block diagram of this design is shown in Figure 2. The sensing front-end consists of a precision bias circuit and a bipolar core. The precision bias circuit provides a biasing current for the substrate PNPs of the bipolar core. The outputs of the bipolar core are $V_{\mathrm{BE}1}$ and $V_{\mathrm{BE}2}$. $\Delta V_{\mathrm{BE}}$ and $V_{\mathrm{BE}2}$ are digitized by incremental Δ-Σ ADC. The bit-stream provided by the incremental Δ-Σ ADC is sent to the digital back-end and processed by an off-chip digital decimation filter to achieve $X=\gamma \cdot \Delta V_{\mathrm{BE}}/V_{\mathrm{BE}}$. Finally, the calibration parameter *k* and constant coefficients *A*, *B* can be obtained by fitting between *X* and temperature; thus, $D_{\mathrm{out}\;}$ is determined.

## 3. Sensing Front-End

It is typical for PNPs to be biased via their emitters, and there are nonzero base and emitter resistances. Considering an equivalent emitter resistance $\left(r_{S}\right)$ and its finite current gain (β), $V_{\mathrm{BE}}$ (for the larger biasing current) and $\Delta V_{\mathrm{BE}}$ can be determined as follows [4]:(6)$$V_{\mathrm{BE}}\approx \frac{kT}{q}\cdot \ln\left(\frac{p_{E}\cdot I_{E}}{t_{S}}\right)+\frac{kT}{q}\cdot \ln\left(\frac{\beta }{\beta +1}\right)+p_{E}\cdot r_{S}\cdot I_{E}$$
(7)$$\Delta V_{\mathrm{BE}}\approx \frac{kT}{q}\cdot \ln\left(p_{E}\right)+\frac{kT}{q}\cdot \left(\frac{\Delta \beta }{\beta \cdot (\beta +1)}\right)+r_{S}\cdot \left(p_{E}-1\right)\cdot I_{E}$$
where $I_{E}$ is the emitter current, and $p_{E}$ is the density ratio of the emitter current. Δβ is the difference in β when there are two different biasing currents ($I_{E}$ and $p_{E}\cdot I_{E}$).

As shown in (6), $V_{\mathrm{BE}}$ is dominated by the first term and has a slightly nonlinear characteristic. The nonlinearity or curvature of $V_{\mathrm{BE}}$ is influenced by the temperature dependence of biasing current $I_{E}$, resulting in the sensor’s systematic error. The PTAT/R biasing circuit (Figure 3) used in this work provides a PTAT current rather than a constant current and, hence, decreases the error. Furthermore, the supply-independent current provided improves the accuracy of $V_{\mathrm{BE}}$ [4]. The bias circuit consists of two PNPs, biased at a 1:5 current ratio. An amplifier forces its corresponding $\Delta V_{\mathrm{BE},b}$ in a bias circuit across a biasing resistor ($R_{1}$) to generate the biasing current ($I=\Delta V_{\mathrm{BE},b}/R_{1}$). The current mirror copies the biasing current to the bipolar core.

### 3.1. Effect of Current Gain β

Since a small biasing current ($I_{E}$) decreases the effect of $r_{S}$, the last terms in (6) and (7) can almost be ignored. The spread (up to 50%) and temperature dependence of β influence the accuracy of $V_{\mathrm{BE}}$ [13]. As is the case in the 180 nm CMOS process (β∼2.7), this effect is significant. This problem can be solved by adding a β-compensation resistor ($R_{\beta }=R_{1}/5$) in series with the base of $Q_{2}$ [14]. The generated biasing current can be rewritten as:(8)$$I=\left(\Delta V_{\mathrm{BE}}/R_{1}\right)\cdot \left(\beta +1\right)/\beta$$
where β is the current gain of the $Q_{2}$. Assuming the two PNPs in the bipolar core have the same current gain, their collector current is equivalent to $\Delta V_{\mathrm{BE}}$/$R_{1}$. However, current mirror and PNPs’ mismatch will limit this approach.

Adding a β-compensation resistor corrects the error on $V_{\mathrm{BE}}$ caused by β, but it does not correct the error on $\Delta V_{\mathrm{BE}}$ caused by β. The current ratio of five results in $\Delta V_{\mathrm{BE}}$ having an appropriate temperature sensitivity in addition to a decrease in the current dependence on β. As shown in Figure 4, β varies with current density and the area of PNPs. Choosing suitable current density and area of PNPs to minimize the current dependence in β is important. When the area of PNPs is 5 × 10 μm, the curve of error is smoother than others and converges to zero at higher currents. In order to make the final temperature error caused by the effect of β on $\Delta V_{\mathrm{BE}}$ as little as possible and have smaller variation with the bias current, the value of I is optimized to be 140 nA at 27 °C, and the area of PNPs is chosen to be 5 × 10 μm in the 180 nm CMOS process. At the same time, the power consumption of the analog front-end has been significantly reduced.

### 3.2. Opamp Topology

As shown in Figure 5, an adaptive self-biasing opamp [2] consisting of a PMOS input pair with diode-connected NMOS loads is used in this work. When operating in a close-loop configuration, the amplifier works in a negative feedback loop and stabilizes the circuit. Meanwhile, the output of opamp is connected to the gate of $M_{P10}$. The diode-connected load makes the gate of $M_{P10}$ a low-impedance node. Therefore, the opamp’s bias current can be reduced to meet the easily satisfied load requirements. The close-loop gain of the opamp is over 74 dB as the temperature and corner vary while the open-loop gain is over 90 dB at 27 °C. At 27 °C, the opamp draws only 980 nA.

### 3.3. Precision Issues

One of the significant sources of inaccuracy is the offset of opamp. The inaccuracy required cannot be easily achieved by just changing sizes of transistors and careful layout. To further improve accuracy, the opamp is chopped [2]. As Figure 5 shows, the input of the opamp is chopped, and the switch at the output of the input stage is used to maintain the correct feedback polarity. Another source of inaccuracy is the mismatch in current sources and bipolar devices. In this case, DEM is used in the current mirrors of the bias circuit and bipolar core, respectively, to improve the accuracy of $\Delta V_{\mathrm{BE}}$. Finally, the required accurate 1:5 current ratio is generated.

## 4. Readout Circuit

Incremental Δ-Σ ADC consists of an easy-to-realize one-bit second-order Δ-Σ modulator and an off-chip digital decimation filter. It can achieve high resolution easily at the cost of conversion rate. There are subtle differences between an incremental Δ-Σ ADC and a conventional one [15]. Firstly, the integration capacitors are reset before each conversion, and the initial states of the integrator are determined. In this case, the ADC does not operate continuously and the decimating filter off-chip can be realized more easily. A one-bit second-order incremental Δ-Σ ADC is used in this paper because it can achieve a certain accuracy in a lower sample rate compared with a first-order ADC [16]. The greatly reduced number of samples decreases the conversion time and power consumption of the sensor.

### 4.1. Incremental Δ-Σ ADC with Charge-Balancing Scheme

A different charge-balancing scheme is required in this incremental Δ-Σ ADC while using only two sampling capacitors. Similar to Ref. [11], the ADC digitizes the ratio *X* = 3 ·$\Delta V_{\mathrm{BE}}$/$V_{\mathrm{BE}}$, which varies from 0.12 to 0.48 as the temperature varies from −50 to +150 °C. γ is chosen to be three in order to achieve a relatively large dynamic range with fewer clock cycles, thus speeding up the conversion rate and reducing energy consumption. In addition, it simplifies the control logic. In each cycle of the ADC, the charge integrated changes corresponding to the bit-stream (BS). It is proportional to 3·$\Delta V_{\mathrm{BE}}-V_{\mathrm{BE}2}$ when BS = 1 while proportional to 3·$\Delta V_{\mathrm{BE}}$ when BS = 0. This charge-balancing scheme is equivalent to connecting the ADC’s input to $3\cdot V_{\mathrm{BE}2}$ and then straddling it with two different references $3\cdot V_{\mathrm{BE}1}+V_{\mathrm{BE}2}$ and $3\cdot V_{\mathrm{BE}1}$ [11].

### 4.2. Sampling Scheme in the Incremental Δ-Σ ADC

In order to realize the gain factor γ = 3, the differential input voltage of the ADC applies the following sequence of voltages: +$\Delta V_{\mathrm{BE}}$, −$\Delta V_{\mathrm{BE}}$, +$\Delta V_{\mathrm{BE}}$, $V_{\mathrm{BE}2}$ when BS = 1 and +$\Delta V_{\mathrm{BE}}$, −$\Delta V_{\mathrm{BE}}$, +$\Delta V_{\mathrm{BE}}$, 0 when BS = 0. As shown in Figure 6, a cascade-of-integrators feedforward form (CIFF) structure was used because the feedforward structure reduces the output swing in each stage, so it is easy to meet integrators’ linearity and slewing requirements [17]. The input of the second-order incremental Δ-Σ ADC changes corresponding to the bit-stream. In each cycle of the ADC, the input is 3·$\Delta V_{\mathrm{BE}}$ when BS = 0 and 3 ·$\Delta V_{\mathrm{BE}}$− $V_{\mathrm{BE}2}$ when BS = 1.

The detailed design of ADC is shown in Figure 7, cascoded FIAs are used in both integrators to achieve higher energy efficiency. The overall circuit is designed symmetrically to reduce gain errors due to mismatched capacitors. Chopping is applied only to the first integrator to reduce 1/f noise and offset. The effect caused by first integrator will impose on the input, while the rest will be suppressed by the loop. Since the same capacitors sample $\Delta V_{\mathrm{BE}}$ or $V_{\mathrm{BE}2}$, there is no need for DEM or the associated logic.

As shown in Figure 8, the first-stage integrator uses two cycles of non-overlapping clock phases $\Phi_{1}$ and $\Phi_{2}$ to sample and integrate. During the last phase of the two cycles $\left(\Phi^{\prime \prime }{}_{1}\right)$, the second-stage integrator samples the output voltage of the first-stage integrator and then integrates during the next phase $\left(\Phi^{\prime \prime }{}_{2}\right)$. Then the output voltages of the two stages are summed by an SC-adder and evaluated by the comparator to generate the output bit-stream [16]. The comparator is triggered by $\Phi_{eval}$ at the interval of the non-overlapping clock phases. The sampling capacitor $C_{s1}$ is 600 fF, which is determined by the constraints of kT/C noise. The sampling frequency is set to 100 kHz, so each clock phase takes 5 μs, the cycle required for generating one-bit output is 20 μs. For the proposed incremental Δ-Σ ADC, 512 cycles are enough to obtain a 15-bit resolution, and a complete conversion takes 10.24 ms.

### 4.3. Cascoded FIA

For the Δ-Σ ADC, the core module is the integrator. The traditional Δ-Σ ADC often uses static current-biased OTAs as its integrators, and a bias current generation circuit is also needed to ensure the static operating point of the OTA. The OTAs are typically the most power-hungry blocks due to their high static current [18].

Unlike traditional OTA, FIAs do not need any static bias current. In addition, FIA only works in half of the cycle and draws a dynamic current from VDD during the reset phase. However, a conventional single-stage FIA can only achieve a very modest DC gain (less than 20–30 dB). The two-stage FIA realized in [19,20] increases the DC gain at the cost of incurring an increased input-referred noise. In this paper, a cascoded FIA is used as the integrator in Δ-Σ ADC to improve the DC gain while not affecting the noise. The operating point of the FIA is time-varying, and the average transconductance $G_{m,avg}$ of the FIA determines the settling speed. $G_{m,avg}$ is strongly related to the value of the reservoir capacitor, which is used as a power source. However, it is weakly dependent on the device dimensions and absolute temperature [18].

As shown in Figure 9, FIA consists of a pair of cascoded inverters powered by the reservoir capacitor. It adapts a common source and common gate structure, and the gate of the cascode MOSFETs ($M_{P3,4}$ and $M_{N3,4}$) is connected to VCM = VDD/2 to reduce additional bias voltage. The body and source of the cascode MOSFETs are connected to form low threshold voltage. As a result, the equivalent transconductance of the MOSFETs increases, so the DC gain increases. The cascode MOSFETs operate in weak inversion regions to prevent the main MOSFETs ($M_{P1,2}$ and $M_{N1,2}$) from entering the triode region. Once the reset phase (sample phase $\Phi_{1}$) starts, the reservoir capacitor $C_{RES}$ is pre-charged to VDD and GND. In order not to impact the sample process, the inputs and outputs of the FIA are connected to the common-mode voltage (VCM). During the amplification phase (integrate phase $\Phi_{2}$), the cascoded inverters are powered by $C_{RES}$ and amplify the input signal. The charge stored in $C_{RES}$ becomes depleted as current flows through the circuit, and hence, finally, the inverters shut off [20]. The integrator using only one stage dynamic amplifier circuit does not need a common mode feedback circuit, nor does it have the problem of stability. However, the signal swings are reduced for a given linearity because of the cascode. Owing to the input feedforward path and dynamic range scaling of the integrator outputs, this does not severely limit the incremental Δ-Σ ADC [18]. The reservoir capacitance value of the first stage integrator is scaled to be 5.6 pF to ensure DC gain over 80 dB in a full temperature range. The one in the second-stage integrator is scaled to be 2 pF to ensure DC gain over 60 dB. Chopping at fs/2 is applied only to the first integrator to suppress the effect of 1/f noise and offset. The two integrators of the Δ-Σ ADC draws 4.5 uA from 3.3 V supply in simulation.

### 4.4. Switch Leakage

Leakage current rises with temperature and has a significant effect at high temperatures. As shown in Figure 7, the outputs of the bipolar core($V_{\mathrm{BE}1}$ and $V_{\mathrm{BE}2}$) are directly connected to the sampling network of incremental Δ-Σ ADC. In this case, switch leakage will have an influence on PNP’s bias current; thus, errors will be brought to the sampled voltage. The number of switches is proportionally related to the total amount of leakage. In previous work [11], on-chip calibration is used, and two additional sampling switches are required. This work is calibrated on the digital back-end and only needs six sampling switches. To further reduce switch leakage, T-switches (Figure 10) are used as sampling switches [21]. It consists of two series-connected NMOSFETs and an intermediate voltage-controlling PMOSFET. The PMOSFET biases the central node to $V_{bias}$ (VDD/2 in this case) when the switch is OFF, as one of the two NMOSFETs is always in the deep cut-off region, and T-switches enable a higher accuracy at high temperatures with less switch leakage.

## 5. Exprimental Results

The proposed temperature sensor is realized in a 180 nm CMOS process and occupies 0.13 mm${}^{2}$ (see Figure 11). The sensor includes sensing front-end, incremental Δ-Σ ADC and control logic. At room temperature, the sensing front-end draws a current of 4.5 μA, the comparator draws a current of 2 μA under 3.3 V analog supply, the two-stage integrators draw a current of 7 μA under 3.3 V digital supply, and the control logic draws a current of 630 nA under 1.8 V digital supply; a distribution of power consumption is shown in Figure 12. The supply sensitivity from 2 to 3.8 V is 0.06 °C/V. The calibration logic and decimation filter are realized off-chip for flexibility.

For the temperature sensing characterization, 15 samples packaged in ceramic DIP packages from one batch are measured over the temperature range from −50 to 150 °C using a thermal chamber. The test setup is shown in Figure 13. The dimension of the thermal chamber is about 50 × 51 × 62 cm and divided into three layers, the upper layer, the middle layer and the lower layer. The PCB with test chips is placed in a metal box on the middle layer to decrease the fluctuation of temperature around the test chips. When the indicator of the thermal chamber reaches the required temperature, keep it for more than one hour to stabilize the ambient temperature, and then the measurements are conducted. A reference temperature sensor (Pt-100 resistor) with less than 30 mK inaccuracy is placed in the center of test chips during the measurement. In order to ensure performance at high temperatures, the PCBs with setup signals are placed outside the thermal chamber, including power supplies (LDOs) and onboard MCUs (STM32L011K4), which work as the bridge of communication, etc. High temperature-resistant signal wires connect the PCBs inside and outside of the temperature chamber. The logic analyzer samples the output bit-stream. The 100 kHz system clock is generated by signal generators. In order to verify the main functions of the temperature sensor more accurately, we do not design the system clock generator internally.

According to (4) and (5), *X* = 3·$\Delta V_{\mathrm{BE}1}$/$V_{\mathrm{BE}}$ is digitized from −50 to 150 °C and is fitted with temperature as follows:(9)$$T=A\cdot \left(\frac{k\cdot X}{k\cdot X+1}\right)+B$$
where *A* = 634.7, *k* = 5.052, *B* = −289.5. All the test chips apply the same coefficients, and then the inaccuracy is calculated. As shown in Figure 14, the measured untrimmed inaccuracy is less than ±1.4 °C (3σ). This improves to ±0.8 °C (3σ) (see Figure 15) after one-point calibration at 40 °C, which is a point in the middle of the temperature measurement range. Calibration at this point can keep a balance between the inaccuracies at the minimum of the temperature measurement range and maximum of the temperature measurement range. In this case, the best inaccuracy is achieved after one-point calibration at 40 °C.

Table 1 summarizes the temperature sensor’s main characteristics and compares it with the state-of-the-art. Refs. [9,10,22] are temperature sensors working at high temperature and [17] have similar supply voltages. Although the temperature sensor works at 3.3 V supply, it consumes significantly less power than most other designs. It also has a better inaccuracy than [10,22].

## 6. Conclusions

An energy-efficient BJT-based temperature sensor employing a Δ-Σ ADC as a readout interface is designed and fabricated in a 180-nm standard CMOS process. The cascoded FIA is used as the integrator instead of the conventional OTA to achieve a low power consumption. To enhance the accuracy, chopping, DEM, β-compensation resistor and optimized bias current are applied in the front-end while the charge-balancing scheme and T-switch are used in Δ-Σ ADC. The accuracy of the proposed sensor is ±0.8 °C (3σ) from −50 °C to 150 °C after one point calibration at 40 °C. The total power consumption is 45.7 μW.

## Figures and Tables

**Figure 1 sensors-22-09381-f001:**
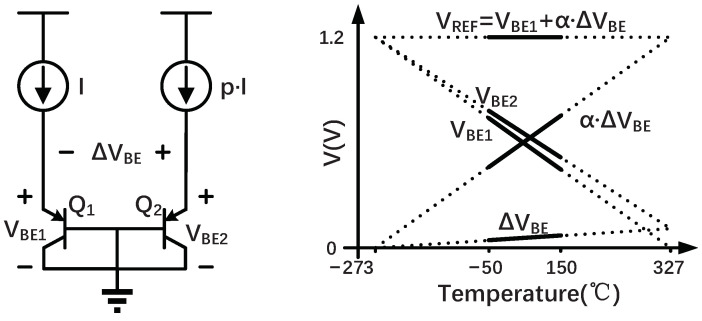
The required voltages generated by two substrate PNPs for temperature measurement.

**Figure 2 sensors-22-09381-f002:**
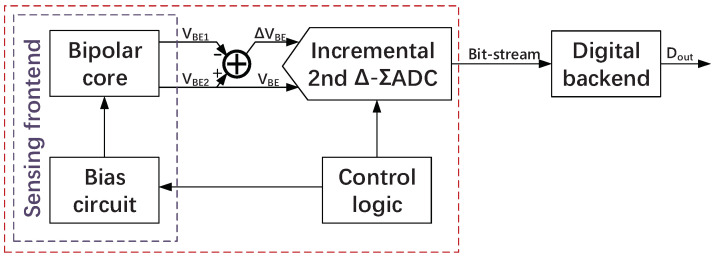
Block diagram of the proposed temperature sensor.

**Figure 3 sensors-22-09381-f003:**
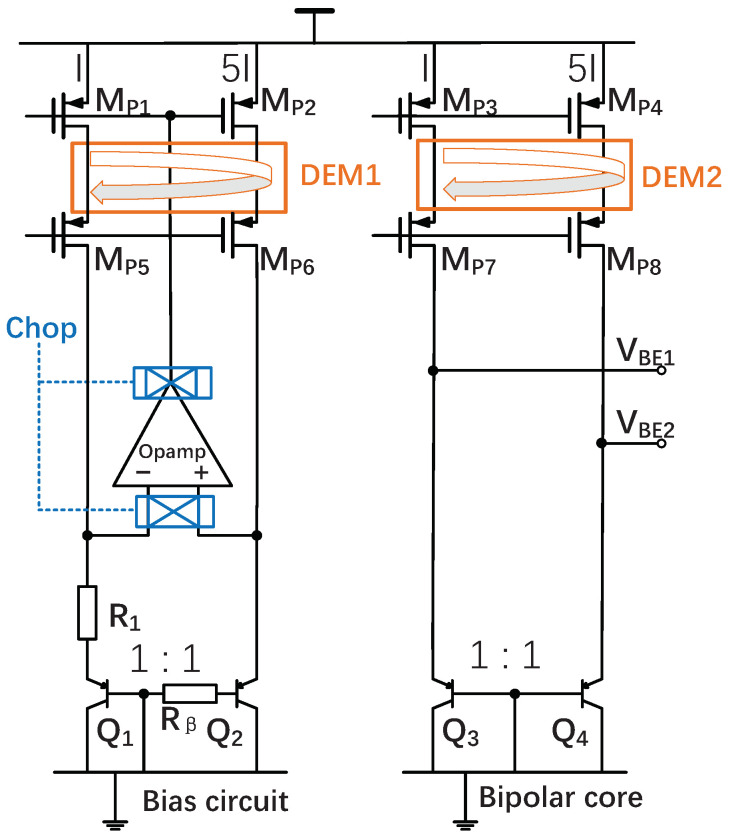
Circuit diagram of the sensing front-end.

**Figure 4 sensors-22-09381-f004:**
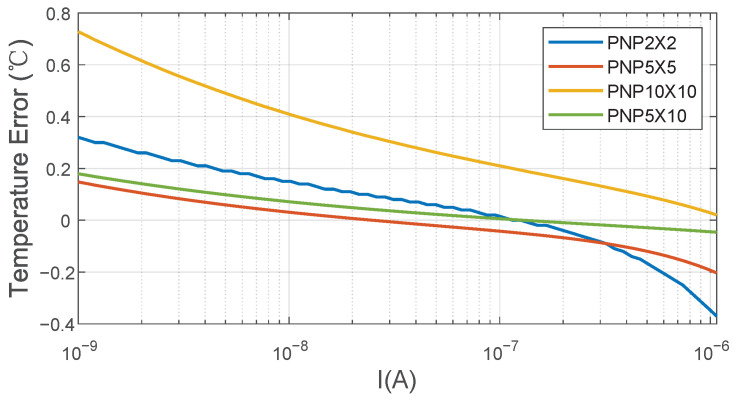
The final temperature error caused by the effect of β on $V_{\mathrm{BE}}$ as a function of biasing current *I* in the 0.18 μm CMOS process.

**Figure 5 sensors-22-09381-f005:**
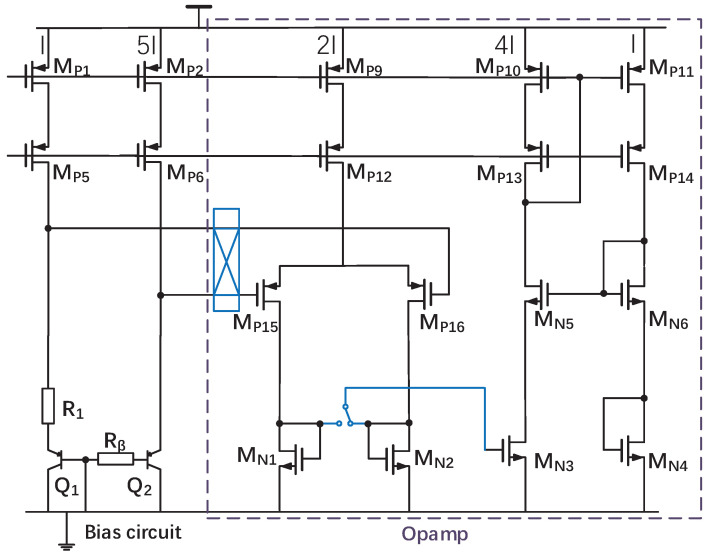
Circuit diagram of opamp with simplified circuit diagram of the bias circuit.

**Figure 6 sensors-22-09381-f006:**
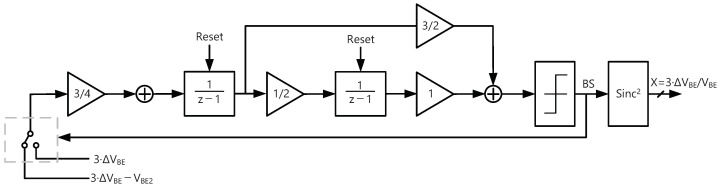
Block diagram of the incremental Δ-Σ ADC.

**Figure 7 sensors-22-09381-f007:**
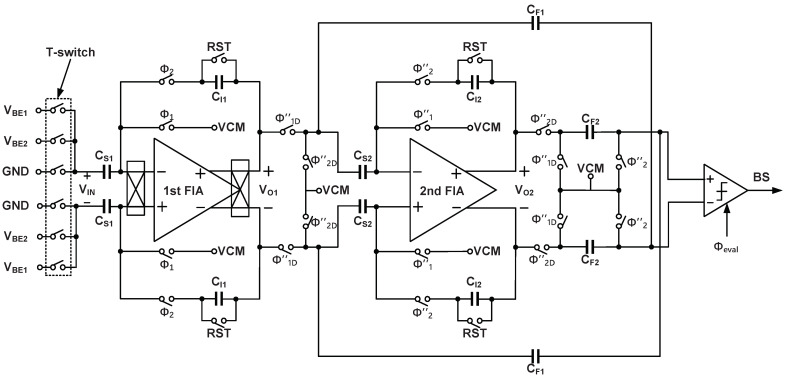
Schematic of the incremental Δ-Σ ADC.

**Figure 8 sensors-22-09381-f008:**
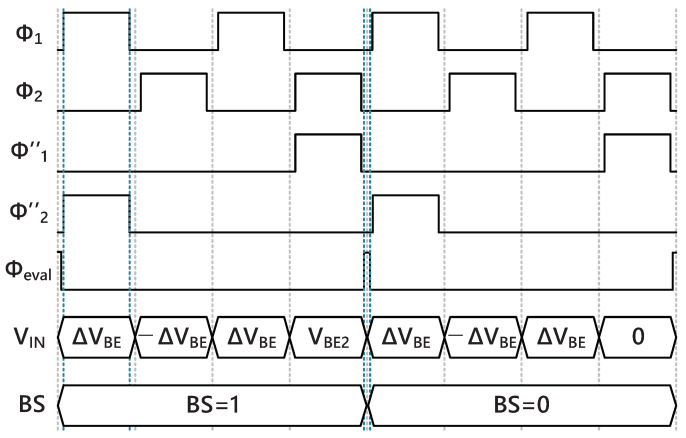
Timing diagram of the incremental Δ-Σ ADC.

**Figure 9 sensors-22-09381-f009:**
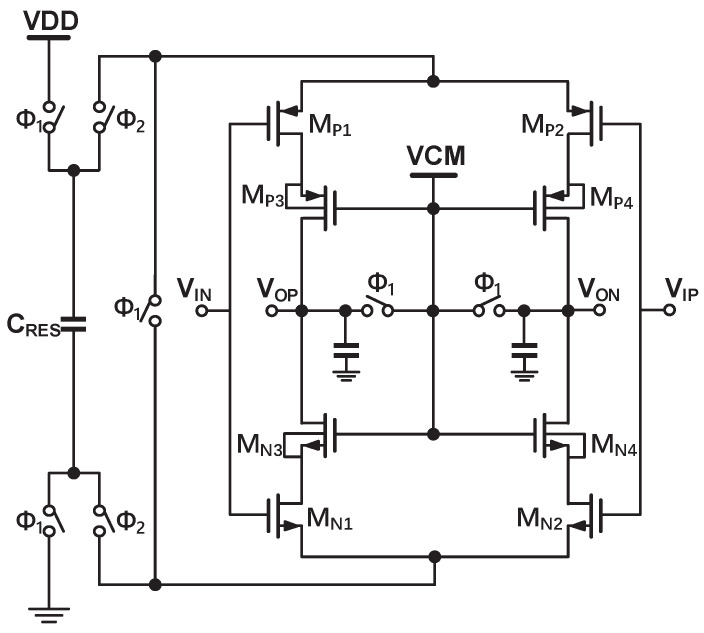
The proposed cascoded FIA.

**Figure 10 sensors-22-09381-f010:**
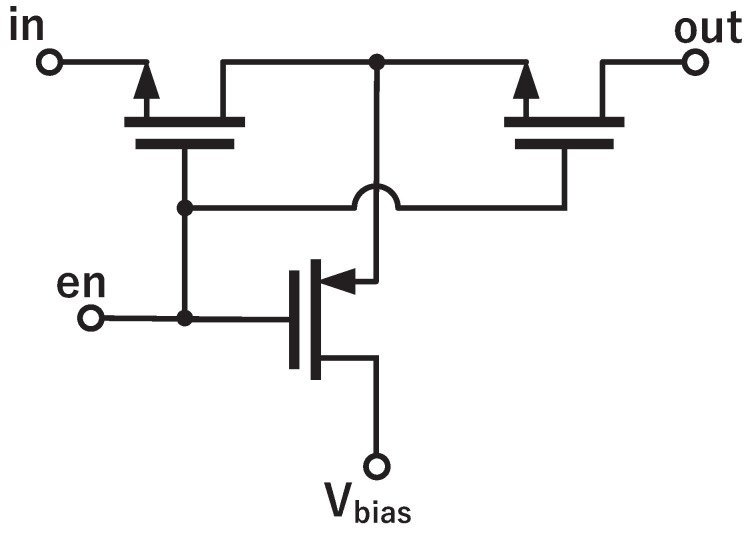
Low-leakage switch.

**Figure 11 sensors-22-09381-f011:**
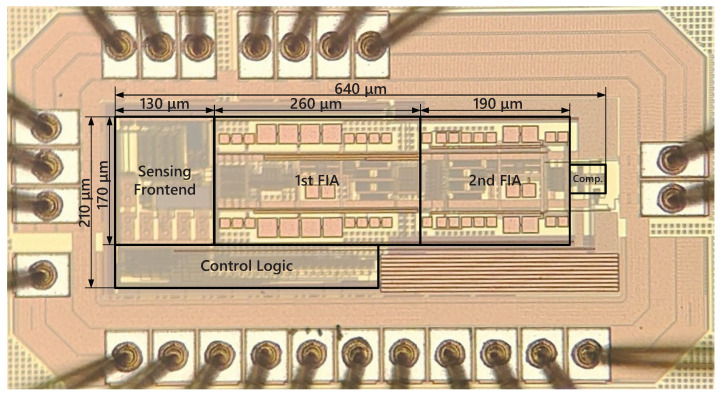
Photomicrograph of the temperature sensor.

**Figure 12 sensors-22-09381-f012:**
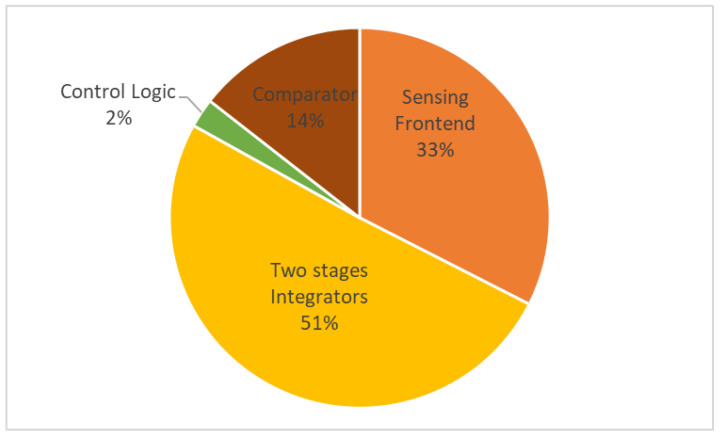
Power consumption distribution.

**Figure 13 sensors-22-09381-f013:**
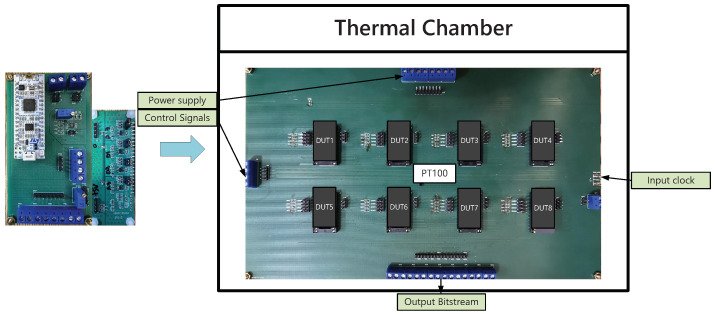
Test setup for the CMOS temperature sensors.

**Figure 14 sensors-22-09381-f014:**
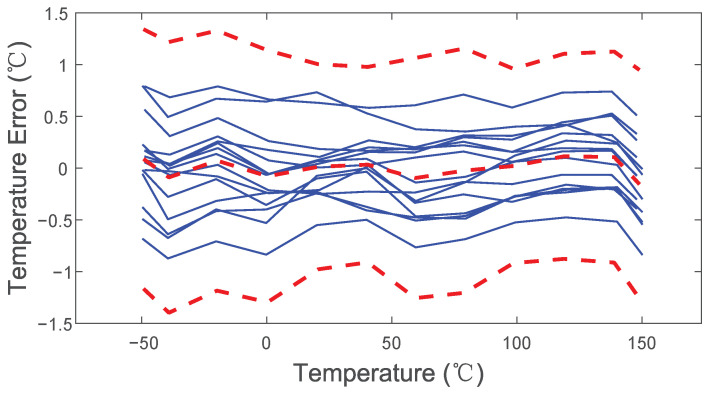
Untrimmed temperature error of 15 sensors; red dashed lines refer to the average and 3σ limits.

**Figure 15 sensors-22-09381-f015:**
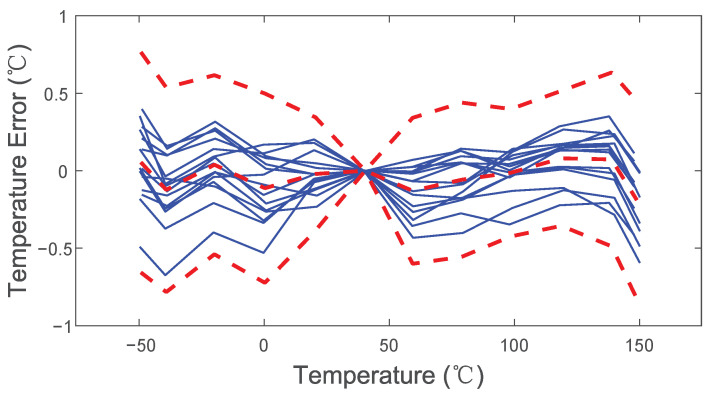
Temperature error of 15 sensors after one-point calibration; red dashed lines refer to the average and 3σ limits.

**Table 1 sensors-22-09381-t001:** Performance summary and comparison with recently published research.

	[9]	[10]	[17]	[22]	This Work
Technology	160 nm CMOS	16 nm FinFET	130 nm CMOS	1 μm CMOS	180 nm CMOS
Area (mm${}^{2}$)	0.1	0.0126	0.29	0.41	0.13
Temp. range (°C)	−55∼200	−50∼150	−40∼125	25∼225	−50∼150
Calibration	1-point	0-point	1-point	1-point	1-point
Supply (V)	1.8	1.8	2–3.6	4.5	1.8/3.3
Power (μW)	39.6	1210	313.5	90	45.7
Inaccuracy (°C)	±0.4 (3σ)	±2 (3σ)	±0.47 (3σ)	±1.6	±0.8 (3σ)
Relative inaccuracy (%) ^1^	0.31	2	0.57	1.6	0.8
Resolution (°C)	0.02	0.38	0.016	0.2	0.04
Conversion rate (ms)	4.2	0.27	5.12	100	10.24
Energy/Conv. (nJ)	150	330	1600	9000	468

^1^ Relative inaccuracy(%) = 100 × (Peak to Peak 3*σ* inaccuracy)/(max temperature-min temperature).

## Data Availability

Not applicable.

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
