# Peer review of "An Energy-Efficient BJT-Based Temperature Sensor with ±0.8 °C (3*σ*) Inaccuracy from −50 to 150 °C"

_sensors, 2022, doi:10.3390/s22239381_

Round 1

Reviewer 1 Report

Obs.1. More information regarding calibration for the sensors, such as the physical layout inside the thermal chamber and the time elapsed until the measurements stabilize, shall be added.
Obs.2. For a temperature domain from -50 C to 180 C the authors decided to one single point of calibration. Why was it not decided to choose at least 3 calibration points (at the minimum of the domain, at maximum of the domain and at a value in the middle of the domain) – independent of 15 sensors experiments.
Obs.3. The reason why the calibration was chosen to be done at 40C and not at 100C, for example, shall be explained.
Obs.4.  The inaccuracy of 0.8% is the same for the all-measured domain temperatures?
Obs.5.  Taking into consideration from your description that Vref is temperature independent (equation 3, pag.3/13) please confirm that the hole measured set-up is not influenced in accuracy by, an eventuality, temperature build-up during operation.
Obs.6. In your case, is the temperature error time dependent in addition to current dependent?
Obs.7. Can you provide some observations, or conclusions, regarding the repeatability of manufacturing of each sensor in relation with deviation in functionality?

Author Response

Response to Reviewer

Dear reviewer,

Thank you for your letter and comments on our manuscript " An Energy-Efficient BJT-Based Temperature Sensor with ±0.8℃(3σ) inaccuracy from -50℃ to 150℃" (ID: sensors-2065720). The comments from you, which significantly improve our manuscript, are greatly appreciated. The manuscript has been revised base on these comments, and the responses to you are listed below. Please see the attachment.

Chuyun Qin, et.al

Reviewer 2 Report

Please read the attached PDF file. 

Author Response

Response to Reviewer 2 Comments

Dear reviewer,

Thank you for your letter and comments on our manuscript " An Energy-Efficient BJT-Based Temperature Sensor with ±0.8℃(3σ) inaccuracy from -50℃ to 150℃" (ID: sensors-2065720). The comments from you, which significantly improve our manuscript, are greatly appreciated. The manuscript has been revised base on these comments, and the responses to you are listed below. Please see the attachment.

Chuyun Qin, et.al

Reviewer 3 Report

The authors presented a new structure of bipolar junction transistors (BJT) based temperature senor where cascoded floating-inverter-based dynamic amplifier was used as integrator and an incremental ∆-Σ ADC readout interface was applied to obtain the testing data. This configuration reduced the power consumption of the sensor unit and meantime maintained a moderate unit area as well as an inaccuracy of ±0.8°C from -50°C to 150°C after one-point calibration.

This result offered an alternative choice for circuits made with standard 180-nm CMOS processes. Yet overall the device shown in Fig. 11 was still very complicated as compared to conventional resistive or diode-based thermal sensors, or thin-film-based thermocouples. This may limits the application potential of this kind of sensors.

Minor corrections such as in lines 34-35 on page 1, “…low leakage SOI processes are used in [9]” and “… 16nm FinFET process is attempted in [10]” might be changed into “in Ref. 9” and “in Ref. 10”, respectively.

Author Response

Response to Reviewer 3 

Dear reviewer,

Thank you for your letter and comments on our manuscript " An Energy-Efficient BJT-Based Temperature Sensor with ±0.8℃(3σ) inaccuracy from -50℃ to 150℃" (ID: sensors-2065720). The comments from you, which significantly improve our manuscript, are greatly appreciated. The manuscript has been revised base on these comments, and the responses to you are listed below.

Chuyun Qin, et.al

Point 1: The authors presented a new structure of bipolar junction transistors (BJT) based temperature senor where cascoded floating-inverter-based dynamic amplifier was used as integrator and an incremental ∆-Σ ADC readout interface was applied to obtain the testing data. This configuration reduced the power consumption of the sensor unit and meantime maintained a moderate unit area as well as an inaccuracy of ±0.8°C from -50°C to 150°C after one-point calibration.

This result offered an alternative choice for circuits made with standard 180-nm CMOS processes. Yet overall the device shown in Fig. 11 was still very complicated as compared to conventional resistive or diode-based thermal sensors, or thin-film-based thermocouples. This may limits the application potential of this kind of sensors.

Minor corrections such as in lines 34-35 on page 1, “…low leakage SOI processes are used in [9]” and “… 16nm FinFET process is attempted in [10]” might be changed into “in Ref. 9” and “in Ref. 10”, respectively.

Response 1:

Thank you for your careful review and recognition of our work. Our work will be further improved to increase application potential. The minor corrections are adapted in the revised manuscript.

Round 2

Reviewer 1 Report

I have no other comments.